# Investigation of the Space Charge and DC Breakdown Behavior of XLPE/α-Al_2_O_3_ Nanocomposites

**DOI:** 10.3390/ma13061333

**Published:** 2020-03-15

**Authors:** Xiangjin Guo, Zhaoliang Xing, Shiyi Zhao, Yingchao Cui, Guochang Li, Yanhui Wei, Qingquan Lei, Chuncheng Hao

**Affiliations:** 1Institute of Advanced Electrical Materials, Qingdao University of Science and Technology, Qingdao 266042, China; Guoaiwei525@163.com (X.G.); zsy19941103@126.com (S.Z.); cc18753214515@163.com (Y.C.); Lgc@qust.edu.cn (G.L.); weiyhui@126.com (Y.W.); leiqingquan@qust.edu.cn (Q.L.); 2State Key Laboratory of Advanced Power Transmission Technology, Changping district, Beijing 102209, China; xingzhaoliang007@163.com

**Keywords:** XLPE/α-Al_2_O_3_ nanocomposites, electrical properties, breakdown strength, electrical conductivity, space charge, pulsed electro-acoustic method

## Abstract

This paper describes the effects of α-Al_2_O_3_ nanosheets on the direct current voltage breakdown strength and space charge accumulation in crosslinked polyethylene/α-Al_2_O_3_ nanocomposites. The α-Al_2_O_3_ nanosheets with a uniform size and high aspect ratio were synthesized, surface-modified, and characterized. The α-Al_2_O_3_ nanosheets were uniformly distributed into a crosslinked polyethylene matrix by mechanical blending and hot-press crosslinking. Direct current breakdown testing, electrical conductivity tests, and measurements of space charge indicated that the addition of α-Al_2_O_3_ nanosheets introduced a large number of deep traps, blocked the charge injection, and decreased the charge carrier mobility, thereby significantly reducing the conductivity (from 3.25 × 10^−13^ S/m to 1.04 × 10^−13^ S/m), improving the direct current breakdown strength (from 220 to 320 kV/mm) and suppressing the space charge accumulation in the crosslinked polyethylene matrix. Besides, the results of direct current breakdown testing and electrical conductivity tests also showed that the surface modification of α-Al_2_O_3_ nanosheets effectively improved the direct current breakdown strength and reduced the conductivity of crosslinked polyethylene/α-Al_2_O_3_ nanocomposites.

## 1. Introduction

Polymer nanocomposites have attracted increasing attention in recent years owing to their improved direct current (DC) dielectric properties and applications as high-voltage DC (HVDC) cable insulation materials. The space charge injected by the electrode and ionized by impurities in the insulation material is liable to accumulate, which leads to partial discharge near the conductor or inside the insulation material, resulting in faster degradation and premature failure of the material. A number of methods have been used to suppress the accumulation of space charge inside the insulation material, including the use of inorganic additives, the melting and blending of polymers, and the graft copolymerization of matrices [1,2,3,4].

Various inorganic nanoparticles, including ZnO, SiO_2_, SiC, TiO_2_, MgO, LSMO, and zeolite, have been used to inhibit space charge accumulation in nanocomposites. The addition of these inorganic nanoparticles suppresses the accumulation of space charge, reduces the electrical conductivity, and increases the breakdown strength of insulating materials to various degrees [5,6,7,8,9,10,11,12,13,14,15]. This has been attributed to the introduction of deep trapping states resulted from the generated interface regions by nanoparticles, which can largely reduce charge carrier mobility. In addition, they may increase the dielectric breakdown strength by changing the carrier trap distribution, and thus the space charge distribution.

To more effectively increase the breakdown strength and suppress the space charge accumulation of crosslinked polyethylene (XLPE), the use of intrinsic insulating Al_2_O_3_ fillers can be considered. Several studies of alumina nanopolymer matrix composites have shown that the addition of Al_2_O_3_ or sandwich-structured Al_2_O_3_/low-density polyethylene nanocomposites significantly increases the number of deep trapping sites in the nanocomposite, markedly impedes charge transport, reduces the DC conductivity, and suppresses the accumulation of space charge [16,17,18]. Compared to other crystalline forms of Al_2_O_3_, α-Al_2_O_3_ possesses many excellent properties, such as high hardness, high dimensional stability, excellent electrical insulation, high dielectric constant, and low dielectric loss [19]. In addition, when α-Al_2_O_3_ has a shape similar to a sheet, it has a blocking effect on the transport of injected charge along the thickness direction and improves the breakdown strength. So, α-Al_2_O_3_ nanosheets were chosen in this work.

However, the compatibility of the nanoparticles and polymer matrix is an essential problem. The surface should be modified by a coupling agent and higher fatty acids or by a grafting polymerization process to reduce the surface energy of the nanoparticles, improve the compatibility of the nanoparticles and matrix, and promote the dispersion of the nanoparticles in the polymer matrix [20,21]. A silane coupling agent can effectively enhance the compatibility of the nanoparticle and polymer matrix; it can react with hydroxyl groups on the surface of nanoparticles, thus coating the surface of nanoparticles with a silane layer, which can prevent the agglomeration of nanoparticles and improve the dispersion of nanoparticles in the polymer matrix (as shown in Figure 1).

Many insulating materials are used in high-voltage direct cables, such as polyvinyl chloride (PVC), polyethylene (PE), crosslinked polyethylene (XLPE), and silicone rubber (SR). Compared with other insulating materials, XLPE insulating materials have the advantages of large transmission power, high voltage level, long life, and low cost. XLPE is a product after the crosslinking of PE, which is a crosslinking process that changes the molecular structure of PE from linear to 3D mesh and converts thermoplastic PE into thermosetting XLPE, thus improving the heat resistance, mechanical properties, and electrical properties of PE. The crosslinking degree of the crosslinked polyethylene by the peroxide crosslinking method is more uniform. So, XLPE was used as the matrix material, and a dicumyl peroxide (DCP) crosslinking agent was used for crosslinking in this work.

In this work, α-Al_2_O_3_ nanosheets were synthesized by a molten salt method, surface-modified by silane coupling agent KH550, and subjected to a variety of characterization tests. The XLPE/α-Al_2_O_3_ nanocomposites were prepared by the thorough mixing of coated α-Al_2_O_3_, low-density polyethylene (LDPE), and a crosslinking agent (dicumyl peroxide; DCP) [22,23,24]. The resulting nanocomposites were press molded at 356 K under a pressure of 16 MPa, yielding two different films with thicknesses of about 150 μm and 300 μm, respectively, which were used for the subsequent DC voltage breakdown testing and measurements of space charge.

## 2. Materials and Methods

### 2.1. Materials

Al_2_(SO_4_)_3_·18H_2_O, NaCl, and KCl were obtained from Tianjin Damao Chemical Reagent Factory. Na_2_CO_3_ was provided by Shanghai Hushi Laboratory Equipment Co. Ltd. Na_3_PO_4_ was purchased from Tianjin Hengxing Chemical Reagent Manufacturing Co. Ltd. TiOSO_4_ was supplied by Shandong Xiya Reagent Co. Ltd. Ethanol was obtained from Tianjin Fuyu Fine Chemical Co. Ltd. The silane coupling agent (KH550) was obtained from Aladdin Industrial Corporation.

### 2.2. Synthesis of α-Al_2_O_3_

The synthesis was performed using the molten salt method [25,26,27]. Solution A and solution B were prepared as follows. First, 33.322 g of Al_2_(SO_4_)_3_·18H_2_O, molten chloride salts (NaCl and KCl in a 6:4 molar ratio), and a very small amount of TiOSO_4_ were dissolved in 90 mL of deionized water under magnetic stirring. Then, the above solution was heated at 353 K for 1 hour. The obtained clear mixed solution was named solution A. Solution B consisted of 50 mL of distilled water, in which Na_2_CO_3_ and Na_3_PO_4_ were dissolved. Solution B was added slowly to solution A under heating and stirring to allow the generated CO_2_ to escape. After 6 h, the synthesized precursor was obtained as a gel comprising Al(OH)_3_ and the molten salt system. The wet gel was dried at 193 K for 3 h. The dried gel was ball-milled in ethanol for 10 h and dried at 353 K. Finally, α-Al_2_O_3_ was obtained by burning in a muffle stove; after which it was dissolved in hot water, filtered, washed, and dried.

The coated α-Al_2_O_3_ was prepared by suspending nanoscale α-Al_2_O_3_ solid powder in an ethanol solution containing a specific amount of C_9_H_23_NO_3_Si (KH550; the amount of KH550 was 3% α-Al_2_O_3_ by weight), followed by heating and stirring at 348 K for 4 h. The resulting mixed solution was collected, filtered, and dried at 373 K for 4 h [28,29,30]. The final product was coated α-Al_2_O_3_.

### 2.3. Characterization of α-Al_2_O_3_

The crystalline phases of coated α-Al_2_O_3_ and uncoated α-Al_2_O_3_ were identified using X-ray diffractometry (XRD; D-MAX 2500/PC, Rigaku, Tokyo, Japan) at 18 kW, 40 kV, and 40 mA with Cu Kα radiation at room temperature. The appearance and distribution of the powder were observed by scanning electron microscopy (SEM; FEI·Nova·Nano·SEM450, Hillsborough, OR, USA). High-resolution photographs and the corresponding electron diffraction patterns of α-Al_2_O_3_ were obtained by high-resolution transmission electron microscopy (HRTEM; FEI·Tecnai·G2·F30, Hillsborough, OR, USA). Fourier transform infrared (FTIR; VERTEX70v, Bruke, London, England) spectra of the resulting nanoparticles were measured from 4000 to 400 cm^−1^.

### 2.4. Preparation of XLPE/α-Al_2_O_3_ Nanocomposites

LDPE (DH0016-2017) was provided by the China National Petroleum Corporation, Daqing, Heilongjiang. The LDPE/α-Al_2_O_3_ composites were prepared by fully blending the coated α-Al_2_O_3_ with 60.00 g of the LDPE insulating material and 1.50 g of DCP in an internal mixer with a torque rheometer (RM-200C, HaPro Electric Co. Ltd., Harbin). The mixing temperature was 393 K, the mixing time was 15 min, and the rotation speed was 60 r/min. The coated α-Al_2_O_3_ mass concentrations in the LDPE/α-Al_2_O_3_ insulating nanocomposites were set to 0.00 wt % (pure insulating layer without coated α-Al_2_O_3_), 0.20 wt %, 0.50 wt %, 1.00 wt %, and 2.00 wt %. Then, the LDPE/α-Al_2_O_3_ composites were melted with a curing press at 393 K, and the LDPE was converted to crosslinked polyethylene (XLPE) by DCP at 16 MPa and 453 K with 15 min. The α-Al_2_O_3_/XLPE specimens were obtained after cooling at 16 MPa and room temperature. The square specimens with a thickness of 0.15 mm and a side length of 11.5 cm were prepared for the high-voltage breakdown testings, and the round specimens with a thickness of 0.3 mm and a diameter of 8 cm were prepared for the DC electric conductivity tests and measurements of space charge.

### 2.5. Electrical Properties Testing of XLPE/α-Al_2_O_3_ Nanocomposites

#### 2.5.1. Voltage Breakdown Testing

A commercial voltage breakdown tester (DDJ-100 kV, Guance Precision Electric Instrument Equipment Co. Ltd., Beijing, China) was used to measure the dielectric breakdown strength of XLPE/α-Al_2_O_3_ [31]. As shown in Figure 2, two spherical copper electrodes with diameters of 200 mm were used as the high-voltage electrode and grounding electrode, with silicone oil as the medium. The function of silicone oil as a medium is to prevent the formation of bubbles so that the final breakdown field strength measured is the breakdown field strength of air. The voltage was boosted at a linear rate of 1 kV/s until electric breakdown occurred, at which time the voltage data were recorded to give the breakdown voltage of the XLPE/α-Al_2_O_3_ composite. This breakdown experiment was carried out at least 10 times for each specimen, and the experimental data were processed and analyzed by the Weibull statistical distribution [32,33].

#### 2.5.2. Pulsed Electro-Acoustic Method

The pulsed electro-acoustic (PEA) method for measuring the space charge in the sample involves applying a high-voltage electric pulse to both ends of the sample through a coupling capacitor so that the distributed space charge in the sample produces tiny vibrations under the action of the electro-acoustic pulse, which are transmitted in the sample in the form of mechanical waves. Finally, the output of the piezoelectric sensor is converted into an electrical signal to obtain the space charge behavior in the insulating material (Figure 3) [34,35,36]. A commercial testing device (HEYI-PEA-PT1, Shanghai HeYi Electric Co. Ltd., Shanghai, China) was used to measure the space charge density of the XLPE/α-Al_2_O_3_ nanocomposites. The applied electric field strengths were 10 kV/mm and 20 kV/mm, and each test lasted for 30 min.

In addition, the DC electrical conductivity of XLPE/α-Al_2_O_3_ nanocomposites was tested. A commercial insulation resistance tester (ZC-90G, Shanghai Taiou Electronic Co. Ltd., Shanghai, China) was used to measure the insulation resistance. The diameter of the electrode used in this test was 50 mm. The data measured by insulation resistanced tester was resistance (*R*_v_). The resistance of each specimen was measured five times and averaged. The resistivity (*ρ*_v_) can be expressed as
$$\rho_{v}=R_{v}\frac{S}{h}$$
where *S* is the area of the electrode and h is the thickness of the specimen. The DC electrical conductivity was the reciprocal of resistivity.

## 3. Results and Discussion

### 3.1. Characterization Results of α-Al_2_O_3_ Nanosheets

The XRD patterns of α-Al_2_O_3_ obtained by the molten salt method and surface-modified α-Al_2_O_3_ were shown in Figure 4. The characteristic shape and high-resolution peaks at 2θ = 35.15°, 43.355°, and 57.496° indicated the high crystalline of α-Al_2_O_3_. Moreover, the whole diffraction peaks were consistent with those of α-Al_2_O_3_ (JCPDS No.46–1212), with a hexagonal structure. The three strong diffraction peaks were at 35.152°, 43.355°, and 57.496°, corresponding to the three crystal surfaces of α-Al_2_O_3_, (104), (113), and (116). Thus, it was verified that the prepared product was α-Al_2_O_3_ with a hexagonal structure and high crystalline. However, except for a slight difference in the intensity of the peaks, there is no significant difference between the coated α-Al_2_O_3_ and uncoated α-Al_2_O_3_ in XRD patterns.

According to the SEM micrograph of α-Al_2_O_3_ nanosheets shown in Figure 5a, the α-Al_2_O_3_ prepared with this method had good uniformity and high aspect ratio. As shown in Figure 5b, the size of the α-Al_2_O_3_ was about 2 to 5 μm, with thickness ranging from 200 to 400 nm. Figure 5c,d show SEM micrographs of the cross-section of XLPE/coated α-Al_2_O_3_ nanocomposites and XLPE/uncoated α-Al_2_O_3_ nanocomposites containing 2.0 wt % α-Al_2_O_3_, which visually indicated that the α-Al_2_O_3_ nanosheets were dispersed evenly in the XLPE/coated α-Al_2_O_3_ nanocomposites, while the α-Al_2_O_3_ nanosheets aggregated in the XLPE/uncoated α-Al_2_O_3_ nanocomposites. Figure 5e,f show the cross-sections of the nanocomposite specimens containing 1 wt % and 2 wt % of coated α-Al_2_O_3_ nanosheets after the liquid nitrogen quenching section. Figure 5e,f can prove that the most of the α-Al_2_O_3_ wafers were laid in the matrix along the thickness direction of the pressed specimen, which perhaps had a blocking effect on the transport of the injected charge along the thickness direction and improved the breakdown strength of crosslinked polyethylene.

TEM images of the α-Al_2_O_3_ prepared by the melting salt method are shown in Figure 6. As shown in Figure 6a, a piece of flaky alumina was tiled on the copper net with a size of about 2 μm. The lower right side of Figure 6a shows the selected area electron diffraction pattern. Through the calculation of crystalline interplanar spacing, the calibration of crystal plane indices, and the determination of the zone axis, it can be concluded that alumina crystal grew along the (300) crystal surface in the direction of thickness and along the (110) crystal surface in the direction of length. Figure 6b shows the edge portion of the flake alumina (above the red dashed line), which had no clear lattice fringe compared with the interior (below the red dashed line). This indicates that the edge portion had an amorphous structure, which was in contrast to the alumina crystal. Figure 6c shows the surface of the α-Al_2_O_3_, which was rough and had many tiny folds that could potentially improve its compatibility with the polymer matrix. HRTEM measurements were conducted to examine the detailed structure of the samples. The clear lattice fringe of the sample indicated its good crystallinity, and the lattice distance was about 0.209 nm, corresponding to the (113) plane with the sharpest peak at 43.355° in the XRD pattern.

Figure 7 shows the FTIR spectra of the coated and uncoated α-Al_2_O_3_ nanopowders. Both samples showed a strong and wide absorption band in the range of 1000–400cm^−1^, which is characteristic of Al-O bonds in alumina. The characteristic absorption peak at 1061.80 cm^−1^ was indicative of Si-O-Si bonds. Notably, a weak peak appeared at 1625.99 cm^−1^ in the IR spectrum of the coated but not the uncoated α-Al_2_O_3_. This weak peak corresponds to the deformation vibration absorption peak of amidogen. The absorption band at 3435.20 cm^−1^ was attributed to surface hydroxyl groups and the stretching vibration of amino groups on the surface of the alumina nanosheets. Compared with uncoated α-Al_2_O_3_, the coated α-Al_2_O_3_ showed a more intense absorption peak at this wavenumber position on account of the surface modification by KH550. In addition, symmetric stretching vibration peaks of C-H bonds appeared at 2854.63 cm^−1^ and 2923.11 cm^−1^, indicating that the silane coupling agent molecules had successfully adsorbed and bonded on the surface of the alumina.

### 3.2. Electrical Properties Testing Results of XLPE/α-Al_2_O_3_ Nanocomposites

Figure 8a shows a comparison of the DC breakdown strengths of pure XLPE and XLPE/α-Al_2_O_3_ nanocomposites containing 0.2 wt %, 0.5 wt %, 1.0 wt %, and 2.0 wt % coated α-Al_2_O_3_. The addition of α-Al_2_O_3_ improved the DC breakdown field strength of the nanocomposites to various degrees. The DC breakdown field strength of pure XLPE was only about 220 kV/mm. In the XLPE/α-Al_2_O_3_ nanocomposites, the DC breakdown field strength increased with increasing α-Al_2_O_3_ content; even an α-Al_2_O_3_ content as low as 0.2 wt % led to a significant increase (to 260 kV/mm). When the content of α-Al_2_O_3_ increased to 1.0 wt %, the DC breakdown field strength underwent a dramatic increase to 320 kV/mm. However, when the α-Al_2_O_3_ content continued to increase to 2.0 wt %, the DC breakdown field strength of the nanocomposite was reduced to 280 kV/mm (lower than for 0.5 wt % and 1.0 wt %, but higher than for 0.2 wt %). Compared with other working results, the breakdown field strength (320 kV/mm) of the XLPE/coated α-Al_2_O_3_ nanocomposites containing 1.0 wt % coated α-Al_2_O_3_ in this work was much higher than the breakdown field strength (200 kV/mm) of sandwich-structured Al_2_O_3_-LDPE/LDPE/Al_2_O_3_-LDPE nanocomposites in ref 16. Furthermore, it was lower than the breakdown strength (450 kV/mm) of the polyethylene/alumina nanocomposites in ref 17. However, the breakdown strength of the neat LDPE in ref 17 had already reached 450 kV/mm and the breakdown strength of the pure XLPE in this study was only 220 kV/mm. Therefore, the effect of adding coated α-Al_2_O_3_ nanosheets in this work was more significant.

The DC breakdown strengths of nanocomposites containing uncoated α-Al_2_O_3_ nanosheets were also measured. Figure 8b shows a comparision of DC breakdown strengths of the pure XLPE and XLPE/uncoated α-Al_2_O_3_ nanocomposites containing 0.2 wt %, 0.5 wt %, 1.0 wt %, and 2.0 wt % uncoated α-Al_2_O_3_. It can be found by comparing with Figure 8a that the addition of uncoated α-Al_2_O_3_ also improved the DC breakdown strength of XLPE. The changing rules of the uncoated α-Al_2_O_3_ and the coated α-Al_2_O_3_ are basically the same. However, the DC breakdown strengths of XLPE/uncoated α-Al_2_O_3_ are lower in value than the coated α-Al_2_O_3_. When the content of α-Al_2_O_3_ was 1 wt %, the DC breakdown strength of XLPE/coated α-Al_2_O_3_ nanocomposites reached 320 kV/mm, while the DC breakdown strength of XLPE/uncoated α-Al_2_O_3_ nanocomposites was only 280 kV/mm. This illustrated that the surface modification of α-Al_2_O_3_ can improve the DC breakdown strengths of XLPE/α-Al_2_O_3_ nanocomposites, because the surface modification of α-Al_2_O_3_ can effectively prevent the agglomeration of nanosheets and improve the dispersion of nanosheets in the XLPE matrix, thus improving the electrical properties of the nanocomposites.

Why can coated α-Al_2_O_3_ nanosheets greatly increase the DC breakdown strength of nanocomposites? This may have been because the insulating coated α-Al_2_O_3_ nanosheets were intrinsic electrical insulating nanopaticles [37]. Secondly, the redistribution of space charge in the insulation material causes electric field homogenization and decreases the free volume of XLPE [38]. More importantly, flaky Al_2_O_3_ has a blocking effect on the transport of injected charge along the thickness direction just as shown in Figure 9; the charge transport through the flakes becomes inhibited with a detour effect. Therefore, the effect of α-Al_2_O_3_ on the breakdown performance of XLPE is related to the addition of modified α-Al_2_O_3_ nanosheets; up to a certain amount of α-Al_2_O_3_ (1 wt %), the breakdown field increases with the increasing α-Al_2_O_3_ content. When the amount of α-Al_2_O_3_ reaches a certain level (2 wt %), the breakdown field strength of the XLPE decreases rapidly. When the concentration is high, such as at 2 wt %, it is difficult to make the nanoparticles disperse well in the polymer matrix, thus affecting the breakdown performance.

The hypothesis can be confirmed by the DC electrical conductivity of XLPE/α-Al_2_O_3_ nanocomposites. Figure 10 shows the relationship between the content of uncoated and coated α-Al_2_O_3_ and DC electrical conductivity.

As is shown in Figure 10, the DC electrical conductivity decreased with increasing α-Al_2_O_3_ content. When the content of coated α-Al_2_O_3_ reached 1 wt %, the electrical conductivity reached the minimum value (1.043 × 10^−13^ S/m), but when the content continued to increase to 2 wt %, the conductivity increased to 1.312 × 10^−13^ S/m. The same was true for XLPE/uncoated α-Al_2_O_3_ nanocomposites. However, it can be seen that the coated α-Al_2_O_3_ reduced the electrical conductivity of the nanocomposites more than uncoated α-Al_2_O_3_. This can illstrate that the surface modification of α-Al_2_O_3_ can reduce the electrical conductivity of XLPE/α-Al_2_O_3_ nanocomposites. The results of this testing corresponding to the results of the voltage breakdown testing, proving that the XLPE/α-Al_2_O_3_ nanocomposites have the lowest DC electrical conductivity and the highest DC breakdown strength when the content of α-Al_2_O_3_ is 1 wt %.

Under the high voltage, the XLPE will break down and change from an insulator to a conductor, but before the breakdown, the XLPE is not an absolutely non-conductive insulator, and a weak current will appear in the material. Since there are usually only a few free electrons in the insulating material, the charges mainly come from the intrinsic ions and impurity particles. The addition of α-Al_2_O_3_ nanosheets introduces a large number of traps; the number of traps in the nanocomposites is much larger than that in the pure XLPE, which reduces the mobility of carriers in the nanocomposites and prevents the injection of charges, thus reducing the conductivity of the nanocomposites.

The space charges of pure XLPE and the XLPE/α-Al_2_O_3_ nanocomposites containing 0.2 wt %, 0.5 wt %, and 1.0 wt % of α-Al_2_O_3_ varied over time under the application of a DC voltage of 20 kV/mm (Figure 11).

As shown in Figure 11a, immediately after the voltage was applied, both electrons and holes are injected from the cathode and anode separately. After 120 s, the heterocharges and homocharges accumulated rapidly near the cathode in XLPE and reach a maximum, which was about 6 C/m³ in contrast to 3 C/m³ around the anode. The amount of space charge near the both electric poles increased with time and rapidly reached a maximum. At the same time, a small amount of space charge packets developed within the specimen, and the same was true for the XLPE/α-Al_2_O_3_ nanocomposites. However, compared with pure XLPE, the space charge densities near the anode and cathode of XLPE/α-Al_2_O_3_ nanocomposites were markedly reduced. This was particularly notable in the XLPE/α-Al_2_O_3_ nanocomposites containing 0.2 wt % of α-Al_2_O_3_, in which the space charge density near the cathode was only about 3 C/m³, which was half that observed for pure XLPE. 

Figure 12 shows a comparison of the quantity of space charge of pure XLPE and XLPE/α-Al_2_O_3_ nanocomposites containing 0.2 wt % of α-Al_2_O_3_. As shown in Figure 12, the space charge quantity of nanocomposites is significantly lower than that of XLPE. Based on the space charge data for all samples, it appears that the certain addition of α-Al_2_O_3_ can suppress the accumulation of space charge effectively, especially in the case of the XLPE/α-Al_2_O_3_ nanocomposites containing 0.2 wt % of α-Al_2_O_3_.

The space charge densities of pure XLPE and XLPE/α-Al_2_O_3_ nanocomposites containing 0.2 wt % and 0.5 wt % of α-Al_2_O_3_, and their variation with field intensity, are shown in the inset of Figure 13. As shown in Figure 13a, as the intensity of the electric field increased from 10 to 20 kV/mm, the heterocharges ionized by impurities inside the pure sample accumulated near the anode, and a mass of homocharges accumulated near the cathode. When the electric field intensity reached 30 kV/mm, a mass of heterocharges accumulated near the anode of the sample; more importantly, a large quantity of space charges accumulated inside the specimen, with a density of about 3–5 C/mm. Compared with the pure XLPE, the accumulation of space charge in XLPE/α-Al_2_O_3_ nanocomposites showed little variation with field intensity, especially in the case of the XLPE/α-Al_2_O_3_ nanocomposites containing 0.2 wt % and 0.5 wt % of α-Al_2_O_3_. As shown in Figure 13b,c, the addition of α-Al_2_O_3_ effectively inhibited the accumulation of heterocharges and homocharges near the cathode, and space charge accumulation inside the specimen.

These space charge data obtained by the pulsed electro-acoustic method indicate that the addition of α-Al_2_O_3_ suppresses the space charge accumulation in the XLPE. The reason is that the α-Al_2_O_3_ nanosheets introduce a certain number of deep traps in the interface between the XLPE matrix and the nanopaticles. Under the action of an electric field, these deep traps capture the electrons (or holes) injected from the anode (or cathide) and reduce the movable charge inside the insulation material, thus inhibiting the injection of space charge and effectively inhibiting its accumulation in the material.

## 4. Conclusions

The above characterization of α-Al_2_O_3_ nanosheets and the experimental voltage breakdown testing, DC electrical conductivity tests, and space charge measurements demonstrate that α-Al_2_O_3_ synthesized by the molten salt method and surface-modified by KH550 can be evenly dispersed into an XLPE matrix, significantly improving the DC breakdown strength, reducing the electrical conductivity of the polyethylene matrix, and suppressing the accumulation of space charge. In all the samples, XLPE/α-Al_2_O_3_ nanocomposites with a concentration of 1 wt % showed the maximum breakdown field strength and minimum electrical conductivity, and those with a concentration of 0.2 wt % of α-Al_2_O_3_ most effectively inhibited the accumulation of space charge. This was because the interface region generated by the addition of α-Al_2_O_3_ nanosheets successfully introduces a large number of deep traps, which greatly decrease the charge carrier mobility. These results may have implications for the development of materials for HVDC cable insulation.

## Figures and Tables

**Figure 1 materials-13-01333-f001:**
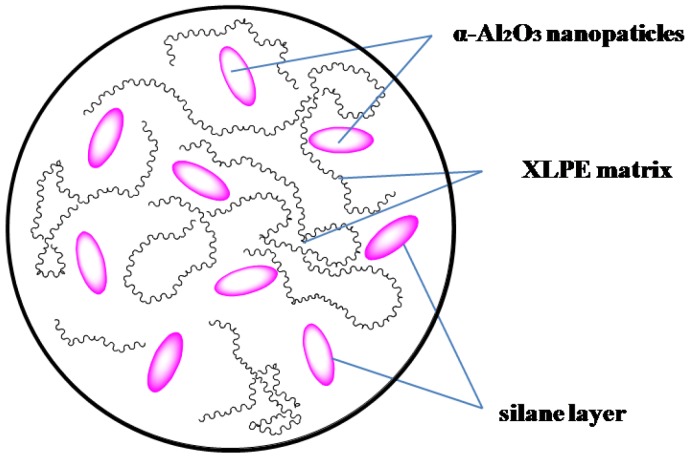
Schematic diagram of dispersion of nanoparticles in matrix.

**Figure 2 materials-13-01333-f002:**
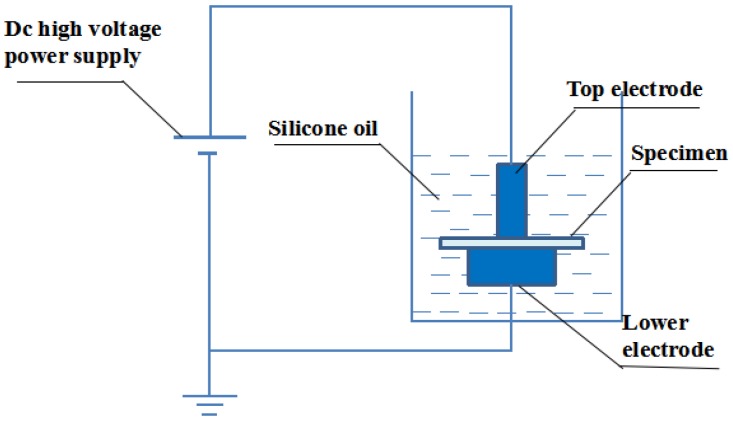
Schematic diagram of the breakdown strength test system.

**Figure 3 materials-13-01333-f003:**
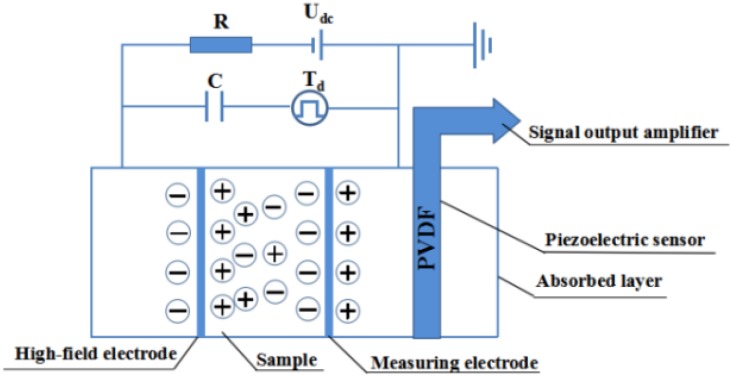
Schematic diagram of pulsed electro-acoustic method.

**Figure 4 materials-13-01333-f004:**
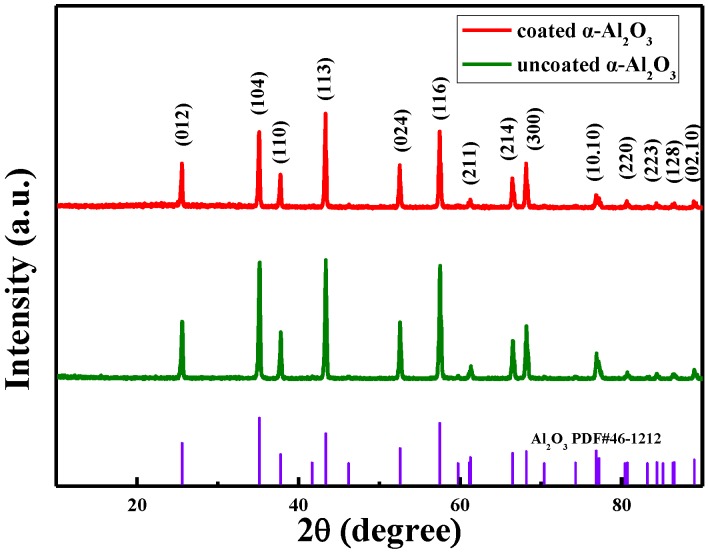
X-ray diffractometry (XRD) patterns of coated and uncoated α-Al_2_O_3_.

**Figure 5 materials-13-01333-f005:**
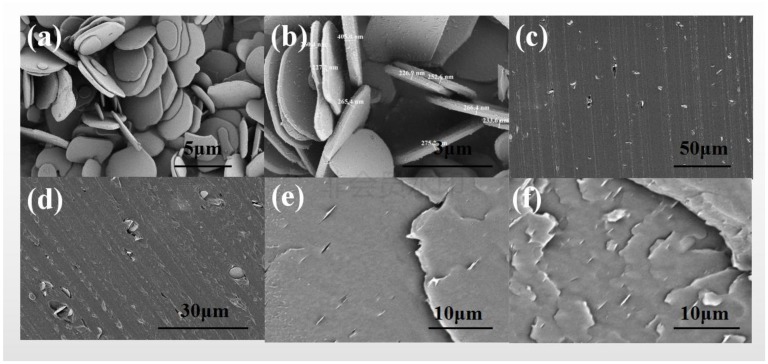
SEM images of (**a**,**b**) uncoated α-Al_2_O_3_ nanosheets, (**c**,**d**) the cross-section of crosslinked polyethylene (XLPE)/coated α-Al_2_O_3_ nanocomposites and XLPE/uncoated α-Al_2_O_3_ nanocomposites containing 2.0 wt % α-Al_2_O_3_ nanosheets and (**e**,**f**) the cross-sections of the nanocomposite specimens containing 1.0 wt % and 2.0 wt% α-Al_2_O_3_ nanosheets after the liquid nitrogen quenching section.

**Figure 6 materials-13-01333-f006:**
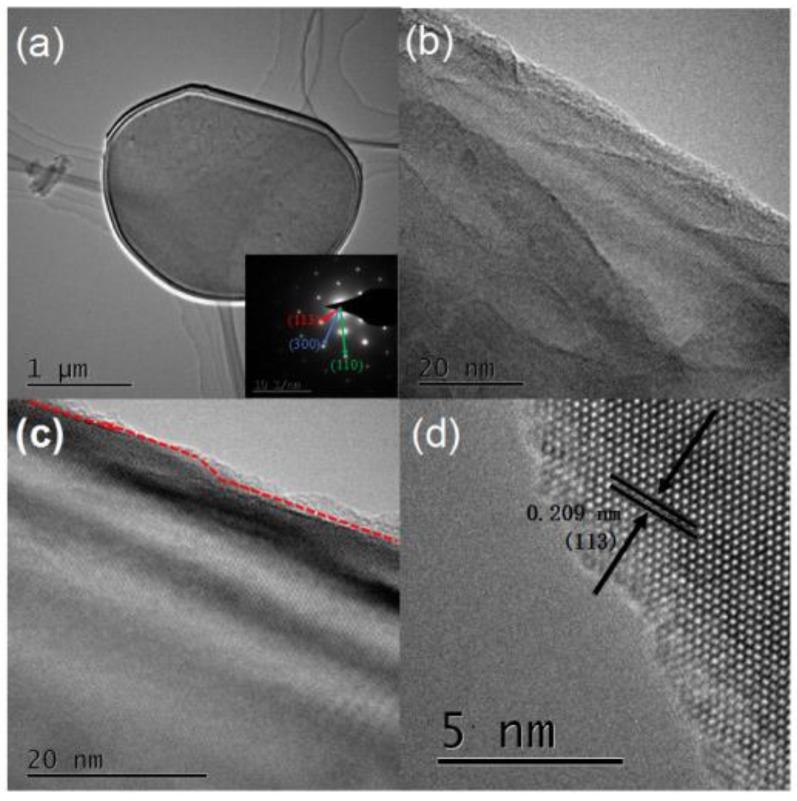
(**a**,**b**) TEM images of uncoated α-Al_2_O_3_. (**c**,**d**) HRTEM images of α-Al_2_O_3_.

**Figure 7 materials-13-01333-f007:**
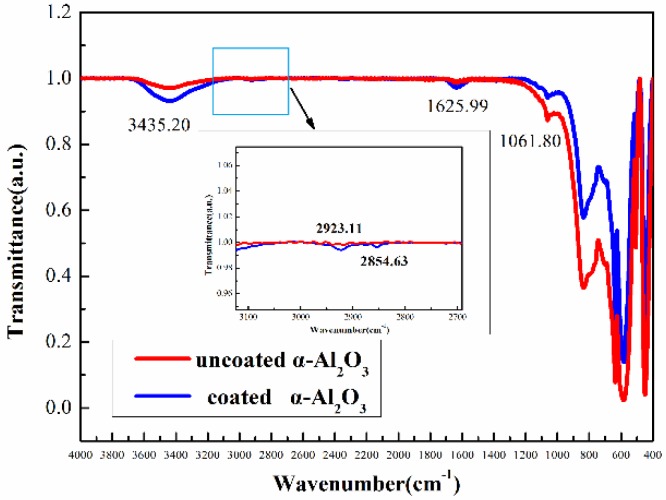
Fourier transform infrared (FTIR) spectrum of coated and uncoated α-Al_2_O_3_.

**Figure 8 materials-13-01333-f008:**
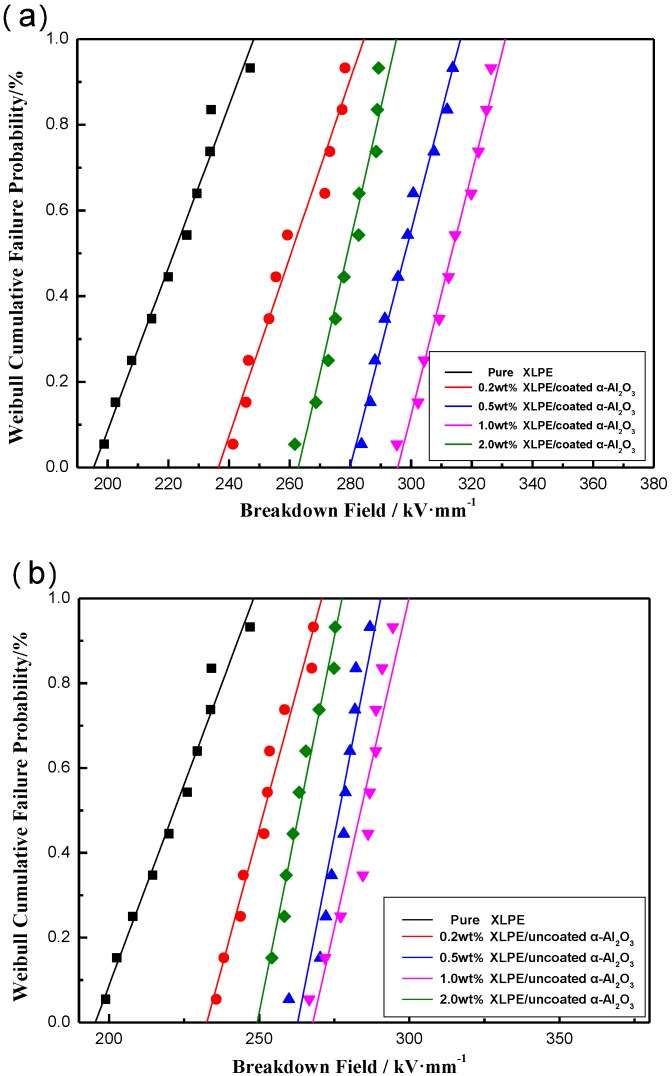
(**a**) Weibull plots comparing the direct current (DC) breakdown strength of pure XLPE and XLPE/α-Al_2_O_3_ nanocomposites containing 0.2 wt %, 0.5 wt %, 1.0 wt %, and 2.0 wt % coated α-Al_2_O_3_ and (**b**) Weibull plots comparing the DC breakdown strength of pure XLPE and XLPE/α-Al_2_O_3_ nanocomposites containing 0.2 wt %, 0.5 wt %, 1.0 wt %, and 2.0 wt % uncoated α-Al_2_O_3_.

**Figure 9 materials-13-01333-f009:**
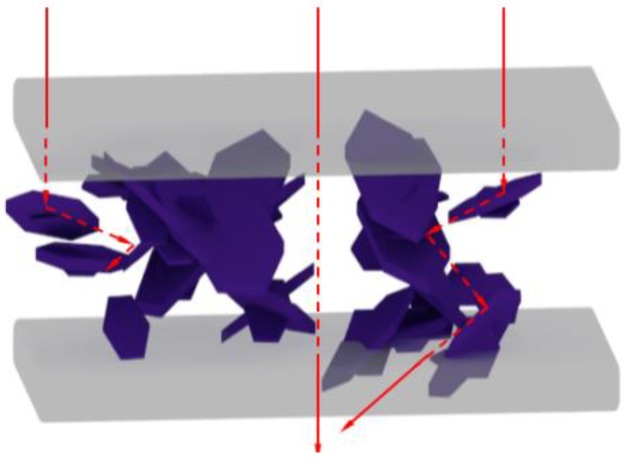
Schematic diagram of the barrier of α-Al_2_O_3_ to injected charge in the XLPE matrix.

**Figure 10 materials-13-01333-f010:**
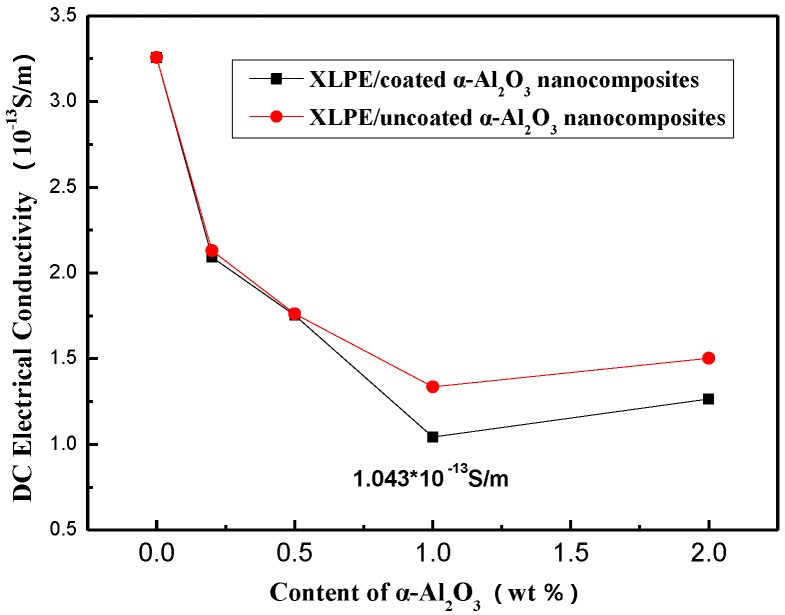
The relationship of the coated α-Al_2_O_3_ content and DC electrical conductivity of XLPE/coated α-Al_2_O_3_ nanocomposites and XLPE/uncoated α-Al_2_O_3_ nanocomposites.

**Figure 11 materials-13-01333-f011:**
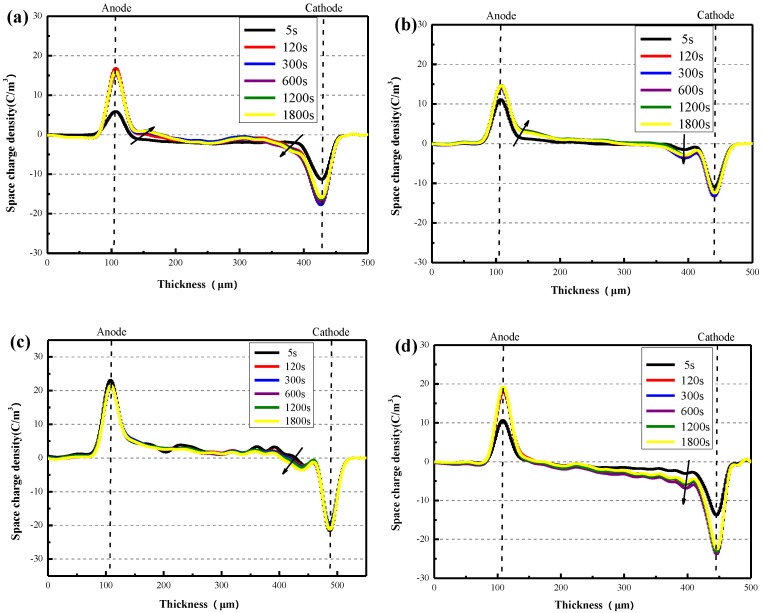
(**a**) The space charge density of pure XLPE varies with the application time of DC voltage (20 kV/mm), (**b**–**d**) The space charge density of the XLPE/α-Al_2_O_3_ nanocomposites containing 0.2 wt %, 0.5 wt %, and 1.0 wt % α-Al_2_O_3_ varies with the application time of DC voltage (25 kV/mm).

**Figure 12 materials-13-01333-f012:**
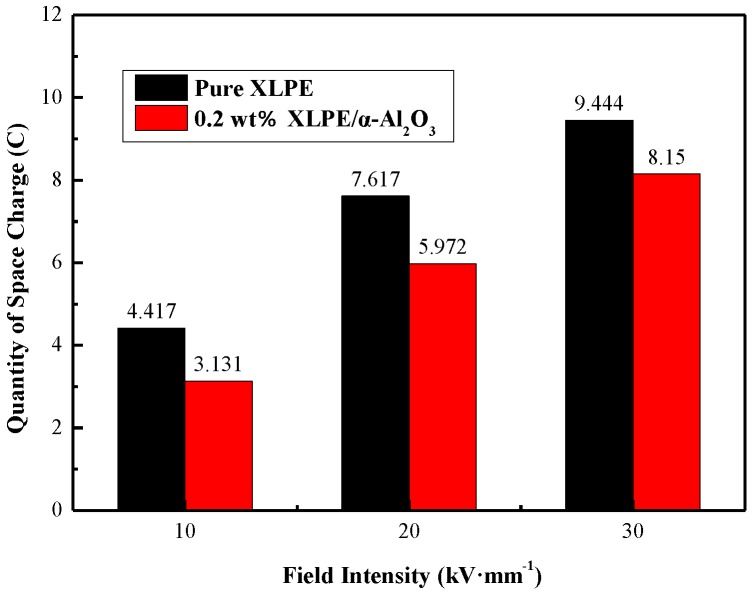
Quantity of space charge of pure XLPE and 0.2 wt % XLPE/α–Al_2_O_3_ nanocomposites.

**Figure 13 materials-13-01333-f013:**
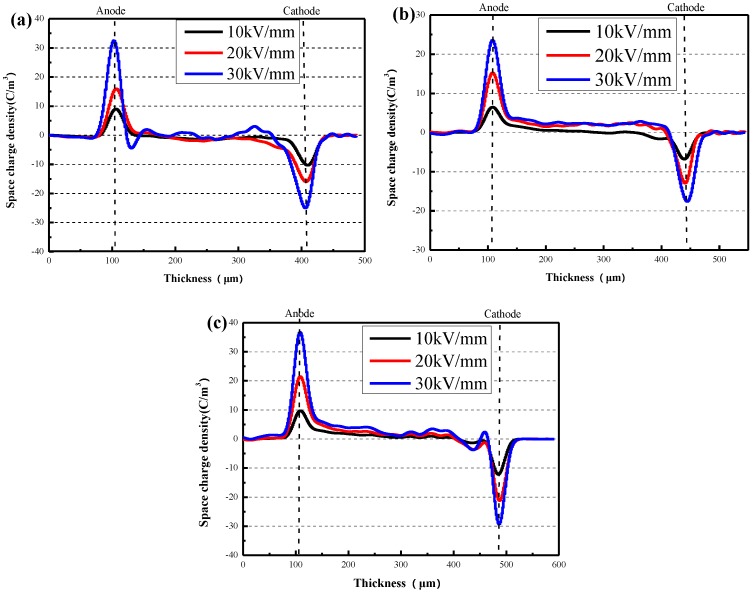
(**a**) The space charge density of pure XLPE varies with field intensity. (**b**,**c**) The space charge density of the XLPE/α-Al_2_O_3_ nanocomposites containing 0.2 wt % and 0.5 wt % of α-Al_2_O_3_ varies with field intensity.

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
