# Peer review of "Investigation of the Space Charge and DC Breakdown Behavior of XLPE/α-Al2O3 Nanocomposites"

_materials, 2020, doi:10.3390/ma13061333_

Round 1
Reviewer 1 Report
The results achieved in the manuscript are of certain interest, but it is not clear what is a real novelty of the work. It is clearly stated in the Introduction that Al2O3 fillers have been previously used as the “charge-breaking” fillers in polyethylene. So, is the novelty here usage of alpha- instead other Al2O3? Or crosslinking of PE? Or surface modification of Al2O3? It should be stated more soundly. Furthermore, the manuscript suffers from lack of comparison with the Al2O3 nanoparticles-filled PE without a crosslinking, without the Al2O3 nanoparticles surface modification or with not alpha- but other modification of Al2O3. A brief example from Characterization section – many SEM images of uncoated Al2O3 nanoparticles are shown, but none for the coated one which were used in the experiments. The same implies for TEM and HRTEM. Furthermore, the achieved results must be compared with “charge-break” efficiency of other PE/Al2O3 materials cited in the Introduction to point out the significance of the work.
Besides the above-mentioned lack, the text should be thoroughly checked to remove some minor errors and typos, e.g. confusing placing of "(b and c)" into a sentence referring to (a) in the caption of Figure 12.
The manuscript can be considered for publication not before addressing the above mentioned comments.
Author Response
Point 1: The results achieved in the manuscript are of certain interest, but it is not clear what is a real novelty of the work. It is clearly stated in the Introduction that Al2O3 fillers have been previously used as the “charge-breaking” fillers in polyethylene. So, is the novelty here usage of alpha- instead other Al2O3? Or crosslinking of PE? Or surface modification of Al2O3? It should be stated more soundly.
Response 1: Sorry,I failed to convey the novelty clearly of the work in the abstract of the work. And I am too detailed about some minor job descriptions, such as the the crosslinking process of polyethylene. The novelty of the work were the usage of surface-modified α-Al2O3 nanosheets instead of other Al2O3 and other morphologies of α-Al2O3. To solve this problem, I have revised the Abstract, Introduction and Materials and Methods, deleted some unnecessary parts, and added some explanations.
Point 2: Furthermore, the manuscript suffers from lack of comparison with the Al2O3 nanoparticles-filled PE without a crosslinking, without the Al2O3 nanoparticles surface modification or with not alpha- but other modification of Al2O3. A brief example from Characterization section – many SEM images of uncoated Al2O3 nanoparticles are shown, but none for the coated one which were used in the experiments. The same implies for TEM and HRTEM.
Response 2: There is no comparison between the electrical properties of crosslinked polyethylene (XLPE) and polyethylene (PE) in the article. Because we chosen the XLPE as the polymer matrix in this work instead of PE. XLPE is a product after crosslinking of PE, crosslinking process which changes the molecular structure of PE from linear to 3d mesh, converts thermoplastic PE into thermosetting XLPE, thus improve the heat resistance, mechanical properties, and electrical properties of PE. I have added their descriptions to the article.
For the lack of comparision of electrical properties between XLPE/coated α-Al2O3 nanocomposites and XLPE/uncoated α-Al2O3 nanocomposites in the work. After your correction, I realized that this is a necessary work. So I spent some time preparing XLPE/uncoated α-Al2O3 nanocomposites and doing voltage breakdown experiments. Then, by comparing their experimental results, I obtained the effect of surface modification of α-Al2O3 nanosheets on the breakdown properties of nanocomposites. The Surface modification of α-Al2O3 nanosheets can improve the DC breakdown strengths and breakdown performance stability of XLPE/α-Al2O3 nanocomposites indeed. I have added these experimental data and discussions to article.
Point 3: Furthermore, the achieved results must be compared with “charge-break” efficiency of other PE/Al2O3 materials cited in the Introduction to point out the significance of the work.
Response 3: As for electrical properties of nanocomposites, I just can comparing the DC breakdown strengths of the XLPE/α-Al2O3 nanocomposites in this work with other works in the references and I have down. But about PEA results, I have no way to compare. Because, due to differences in the size and thickness of experimental specimens and subjective measurement methods, the experimental results obtained by different operators on different instruments vary greatly. Therefore, only the nanocomposites with different α-Al2O3 content can be compared in the work.
Point 4: Besides the above-mentioned lack, the text should be thoroughly checked to remove some minor errors and typos, e.g. confusing placing of "(b and c)" into a sentence referring to (a) in the caption of Figure 12.
Response 4: I have gone over the full text carefully and resolved the suchlike minor errors.
Reviewer 2 Report
The DC current voltage breakdown strength and space charge accumulation in crosslinked polyethylene/α-Al2O3 composites for applications as high-voltage DC (HVDC) insulation materials are analysed in this study.
The study brings some new results on the DC current voltage breakdown strength and space charge accumulation in melt-mixed polymer composites with inorganic fillers. However, I find that some parts of the discussion are not very convincing and therefore this manuscript must be improved before publishing.
Specifically, the advantages of using XLPE instead of HDPE should be better addressed. I recommend also to make DC electrical conductive tests in the composites to see really if there is any electrical percolation threshold between 1 and 2 % wt, as the authors claim. I also recommend to compare the results of composites with coated and uncoated α-Al2O3 particles to see if really that coating is so important in the final properties.
I leave some other questions and comments that I´d like to be responded by the authors.
IntroductionThe authors state the following:
“Compared to other crystalline forms of α-Al2O3, α-Al2O3 possesses many excellent properties, such as high hardness, high dimensional stability, excellent electrical insulation, low dielectric constant and dielectric loss”
My question is if the aim of the work is to produce materials with higher breakdown voltage, then they should work with insulating polymers and fillers with high dielectric constants and low dielectric losses.
There is not any reference in the introduction to other similar works with Al2O3, why?
Materials and Methods2.4. Preparation of XLPE/α-Al2O3 nanocomposites
Why do the authors use crosslinking agent, has XLPE any advantage with respect to normal HDPE?
It is not clear the melt mixing and compression molding conditions for the LDPE/α-Al2O3 composite preparation. What is the mixing temperature, mixing time, rotation speed? Then, the samples were again mixing with the crosslink agent? What were the conditions? And at the end the samples are hot-pressed, what was the pressing time?
The description of the method of processing is not clear at all.
What are the dimensions of the samples tested (apart from thicknesses)?
2.5.1. Voltage breakdown testing
The voltage breakdown tester is a commercial unit or is it a homemade device?
2.5.2. Pulsed electro-acoustic method
The space charge is measured by this PEA method with a commercial unit or is it a homemade device?
Results and Discussion3.1. Characterization results of α-Al2O3 nanosheets
What is the main difference between XRD pattern in coated and uncoated particles?
The authors state the following:
“Most of the α-Al2O3 wafers were laid in the matrix along the thickness direction of the pressed specimen,which perhaps have a blocking effect on the transport of injected charge along the thickness direction and improve the breakdown strength of crosslinked polyethylene”.
Why the authors say that the particles are laid in the thickness direction? From SEM images 5e and 5f, we cannot conclude anything about some preference in the position of the particles within the polymer.
3.2. Electrical properties testing results of XLPE/α-Al2O3 nanocomposites
The authors should test also the electrical properties of XLPE and uncoated α-Al2O3 at 0.2, 0.5, 1 and 2 % wt composites to see if the coating produces any relevant improvement.
Fig. 8 shows an increase in breakdown strength from 220 kV/mm for neat XLPE to 320 kV/mm for composites with 1 % wt of α-Al2O3 particles, and then a decrease to 280 kV/mm for composites with 2 % wt of α-Al2O3 particles. The discussion about the cause of this decrease found in composites with 2 % of particles is not clear. Authors say that an electrical percolation threshold around 1 % wt could explain this decrease. This is easy to demonstrate. Authors should make the DC electrical conductivity tests (conductivity as function of % wt) in their samples to confirm their hypothesis.
Author Response
Point 1: Specifically, the advantages of using XLPE instead of HDPE should be better addressed.
Response 1: I have added the ralated content, which introduces the advantages of using XLPE instead of PE or other insulating material.
Point 2: I recommend also to make DC electrical conductive tests in the composites to see really if there is any electrical percolation threshold between 1 and 2 % wt, as the authors claim.
Response 2: I have made the DC electrical conducticity tests of XLPE/α-Al2O3 nanocomposites and the results of this testing corresponding to the results of the voltage breakdown testing, proving that the XLPE/α-Al2O3 nanocomposites have the lowest DC electrical conductivity and the highest DC breakdown strength when the content of α-Al2O3 is 1 wt%. Therefore,It can be confirmed that the 1 wt% was a global optimum in electrical breakdown strength.
Point 3: I also recommend to compare the results of composites with coated and uncoated α-Al2O3 particles to see if really that coating is so important in the final properties.
Response 3: For the lack of comparision of electrical properties between XLPE/coated α-Al2O3 nanocomposites and XLPE/uncoated α-Al2O3 nanocomposites in the work. After your correction, I realized that this is a necessary work. So I spent some time preparing XLPE/uncoated α-Al2O3 nanocomposites and doing voltage breakdown experiments. Then, by comparing their experimental results, I obtained the effect of surface modification of α-Al2O3 nanosheets on the breakdown properties of nanocomposites. The Surface modification of α-Al2O3 nanosheets can improve the DC breakdown strengths and breakdown performance stability of XLPE/α-Al2O3 nanocomposites indeed. I have added these experimental data and discussions to article.
Point 4: Introduction
The authors state the following:
“Compared to other crystalline forms of α-Al2O3, α-Al2O3 possesses many excellent properties, such as high hardness, high dimensional stability, excellent electrical insulation, low dielectric constant and dielectric loss”
My question is if the aim of the work is to produce materials with higher breakdown voltage, then they should work with insulating polymers and fillers with high dielectric constants and low dielectric losses.
There is not any reference in the introduction to other similar works with Al2O3, why?
Response 4: In the introduction there are three references to similar work with Al2O3, respectively are the "Distinctive electrical properties in sandwich-structured Al2O3/low density polyethylene nanocomposites","Preparation, Microstructure and Properties of Polyethylene/Alumina nanocomposites for HVDC insulation" and "Open-circuit Thermally Stimulated Currents in LDPE/Al2O3 Nanocomposite". The addition of Al2O3 effectively suppresses the space charge, increases the volume resistivity, and presents better breakdown strength of composites.
Point 5: 2.4. Preparation of XLPE/α-Al2O3 nanocomposites
Why do the authors use crosslinking agent, has XLPE any advantage with respect to normal HDPE?
It is not clear the melt mixing and compression molding conditions for the LDPE/α-Al2O3 composite preparation. What is the mixing temperature, mixing time, rotation speed? Then, the samples were again mixing with the crosslink agent? What were the conditions? And at the end the samples are hot-pressed, what was the pressing time?
The description of the method of processing is not clear at all.
What are the dimensions of the samples tested (apart from thicknesses)?
Response 5: XLPE is a product after crosslinking of PE, crosslinking process which changes the molecular structure of PE from linear to 3d mesh, converts thermoplastic PE into thermosetting XLPE, thus improve the heat resistance, mechanical properties, and electrical properties of PE. So XLPE was used as the matrix material in this work.
The mixing temperature was 393 K, mixing time was 15 min and rotation speed was 60 r/min. The LDPE was mixing with the crosslinking agent only once in a torque rheometer. The DCP crosslinking agent was used for crosslinking(From PE crosslinked to XLPE) in this work. At the end the samples are hot-pressed, The LDPE was converted to crosslinked polyethylene (XLPE) by DCP at 16 MPa and 453 K with 15 min.
The square specimens with a thickness of 0.15 mm and a side length of 11.5 cm were prepared for the high voltage breakdown testings and the round specimens with a thickness of 0.3 mm and a diameter of 8 cm were prepared for the DC electric conductivity tests and measurements of space charge.
This part of the content has been modified to make it more clear.
Point 6: 2.5.1. Voltage breakdown testing
The voltage breakdown tester is a commercial unit or is it a homemade device?
2.5.2. Pulsed electro-acoustic method
The space charge is measured by this PEA method with a commercial unit or is it a homemade device?
Response 6: The voltage breakdown tester and the PEA testing device were commercial units, as well as insulation resistance tester.
Point 7: 3.1. Characterization results of α-Al2O3 nanosheets
What is the main difference between XRD pattern in coated and uncoated particles?
The authors state the following:
“Most of the α-Al2O3 wafers were laid in the matrix along the thickness direction of the pressed specimen,which perhaps have a blocking effect on the transport of injected charge along the thickness direction and improve the breakdown strength of crosslinked polyethylene”.
Why the authors say that the particles are laid in the thickness direction? From SEM images 5e and 5f, we cannot conclude anything about some preference in the position of the particles within the polymer.
Response 7: Except for a slight difference in the intensity of the peaks,there is no significant difference between the coated α-Al2O3 and uncoated α-Al2O3 in XRD patterns. Most of the α-Al2O3 wafers were laid or inclined in the matrix along the thickness direction of the pressed specimen, because only the longitudinal sections of α-Al2O3 nanosheets we can see in the SEM images 5e and 5f.
Point 8: Electrical properties testing results of XLPE/α-Al2O3 nanocomposites
The authors should test also the electrical properties of XLPE and uncoated α-Al2O3 at 0.2, 0.5, 1 and 2 % wt composites to see if the coating produces any relevant improvement.
Response 8: XLPE/uncoated α-Al2O3 nanocomposites were prepared and the voltage breakdown experiments were tested. Then, by comparing their experimental results, I obtained the effect of surface modification of α-Al2O3 nanosheets on the breakdown properties of nanocomposites. The Surface modification of α-Al2O3 nanosheets can improve the DC breakdown strengths and breakdown performance stability of XLPE/α-Al2O3 nanocomposites indeed. I have added these experimental data and discussions to article.
Point 9: Fig. 8 shows an increase in breakdown strength from 220 kV/mm for neat XLPE to 320 kV/mm for composites with 1 % wt of α-Al2O3 particles, and then a decrease to 280 kV/mm for composites with 2 % wt of α-Al2O3 particles. The discussion about the cause of this decrease found in composites with 2 % of particles is not clear. Authors say that an electrical percolation threshold around 1 % wt could explain this decrease. This is easy to demonstrate. Authors should make the DC electrical conductivity tests (conductivity as function of % wt) in their samples to confirm their hypothesis.
Response 9: I have made the DC electrical conducticity tests of XLPE/α-Al2O3 nanocomposites and the results of this testing corresponding to the results of the voltage breakdown testing, proving that the XLPE/α-Al2O3 nanocomposites have the lowest DC electrical conductivity and the highest DC breakdown strength when the content of α-Al2O3 is 1 wt%. Therefore,It can be confirmed that the 1 wt% was a global optimum in electrical breakdown strength.
Round 2
Reviewer 1 Report
Addition of the necessary experiments with uncoated alpha-Al2O3 raised some question about reproducibility. The point is that pure XLPE shows substantially different patterns of cumulative failure probability - in one case the probability 1 is achieved at approximately 240 kV.mm-1 while in other case (fig. 11) it is achieved at about 260. According to the authors, these differences are significant (when breakdown strength of filled XLPEs are evaluated). Question is, then, how the real effect of Al2O3 nanoparticle coating can be quantified if blank measurements (i.e. pure XLPEs) are so different under the same conditions? What would repeated measurements show? It is very weak to say only “the DC breakdown strengths of XLPE/uncoated alpha-Al2O3 are slightly smaller…” about the results which, according to the authors, are the main novelty of the manuscript. Besides, better dispersivity of coated Al2O3 is still unproved by SEM. It is absolutely not necessary to show fig. 5a-c, instead the uncoated Al2O3 in XLPE must be shown to prove their coagulation. And fig 5e and 5f still fail to tell something about dispersivity of Al2O3. SEM image of some cross section should be provided.
In conclusion – the results still do not unequivocally support the hypothesis that the coating of Al2O3 has significant effect on the insulating properties. The manuscript must be further improved in the above-mentioned way.
Author Response
Point 1: Addition of the necessary experiments with uncoated alpha-Al2O3 raised some question about reproducibility. The point is that pure XLPE shows substantially different patterns of cumulative failure probability - in one case the probability 1 is achieved at approximately 240 kV/mm while in other case (fig. 11) it is achieved at about 260. According to the authors, these differences are significant (when breakdown strength of filled XLPEs are evaluated). Question is, then, how the real effect of Al2O3 nanoparticle coating can be quantified if blank measurements (i.e. pure XLPEs) are so different under the same conditions? What would repeated measurements show? It is very weak to say only “the DC breakdown strengths of XLPE/uncoated alpha-Al2O3 are slightly smaller…” about the results which, according to the authors, are the main novelty of the manuscript.
Response 1:
Due to temperature, humidity and other test conditions at different times and under different environments, the test results of DC breakdown strengths of the same sample are also quite different, which results in the data of pure XLPE samples in the Fig 8 and Fig 11 are different. In order to solve this problem, I conducted the experiments of DC voltage breakdown testing.
This time, I tested all the specimens again, including the one pure XLPE specimen, four specimens of XLPE/uncoated α-Al2O3 nanocomposites containing 0.2 wt%, 0.5 wt%, 1.0 wt%, and 2.0 wt% uncoated α-Al2O3 and four specimens of XLPE/coated α-Al2O3 nanocomposites containing 0.2 wt%, 0.5 wt%, 1.0 wt%, and 2.0 wt% coated α-Al2O3. To improve the reliability of the data, I conducted 15 times of breakdown experiments on each of 9 specimens and chose the middle 10 data among the 15 data according the size to carry out data processing. For comparision, I put the two figures of DC voltage breakdown tests and two sets of date of DC condutivity tests together. Finally, by comparision, the conclusion can be drawn convincingly that the surface modification of α-Al2O3 nanosheets can improve the DC breakdown strengths and reduce the DC conductivity of XLPE/α-Al2O3 nanocomposites.
Point 2: Besides, better dispersivity of coated Al2O3 is still unproved by SEM. It is absolutely not necessary to show fig. 5a-c, instead the uncoated Al2O3 in XLPE must be shown to prove their coagulation. And fig 5e and 5f still fail to tell something about dispersivity of Al2O3. SEM image of some cross section should be provided. In conclusion – the results still do not unequivocally support the hypothesis that the coating of Al2O3 has significant effect on the insulating properties.
Response 2:
I have deleted the extra SEM pictures and added the necessray SEM pictures you mentioned. In the new SEM images, Fig 5a is to show the size and uniformity of the α-Al2O3 nanosheets; Fig 5b is to show the thickness and high aspect ratio of α-Al2O3 nanosheets; Fig 5c and 5d are the images of cross section of XLPE/coated α-Al2O3 nanocomposites and XLPE/uncoated α-Al2O3 nanocomposites, It can be seen that in the XLPE/uncoated α-Al2O3 nanocomposites, α-Al2O3 nanosheets aggregated, while in the XLPE/coated α-Al2O3 nanocomposites, the α-Al2O3 nanosheets were dispersed evenly in the XLPE matrix. Fig 5e and 5f show the cross sections of the nanocomposite specimens after liquid nitrogen quenching section. Owing to the other reviewer requested to add these SEM images to prove that “The paticles are laid in the thickness direction”. And obviously, with these two graphs, this is proved.
Thanks so much for all your helpful comments! If there are any other problems in my article, please raise them and I will seriously correct them.
Thanks!

Reviewer 2 Report
As I requested in my previous revision, the study was completed with more experimental production and testing work in the second version, which confirms that the use of coated or uncoated particles is not as relevant as the authors stated in their first version. The work introduces interesting experimental work in terms of production and characterization, however the discussion of the results and the final strengths of this work with respect to the state of art are not convincing. Based on this, I do not recommend this work for publication.
Some additional questions/ comments for the authors, after reading the last version and responses from the authors are introduced here:
Abstract
Q1: This paragraph is confuse:
“Direct current breakdown testing, electrical conductivity tests and measurements of space charge indicated that the surface modification of α-Al2O3 improved the direct current voltage breakdown strengths of crosslinked polyethylene/α-Al2O3 nanocomposites”.
The authors should introduce the more important specific values of their study.
- Introduction
Q2: The authors do not respond to this comment in their last revision (response 4):
The authors state the following:
“Compared to other crystalline forms of α-Al2O3, α-Al2O3 possesses many excellent properties, such as high hardness, high dimensional stability, excellent electrical insulation, low dielectric constant and dielectric loss”
My question is if the aim of the work is to produce materials with higher breakdown voltage, then they should work with insulating polymers and fillers with high dielectric constants and low dielectric losses.
Q3: The authors should introduce the specific values obtained in the references 16-18 to compare them after with the values obtained in this work. The results obtained in this work are better than the results provided by references 16-18 with particles of Al2O3?
2.5.2. Pulsed electro-acoustic method
Q4: The authors must explain how the DC conductivity is calculated. Why did the authors choose samples with different thicknesses for both analysis? My recommendation is to test whenever possible samples with the same dimensions for electrical analysis.
- Results and Discussion
3.1. Characterization results of α-Al2O3 nanosheets
Q5: The authors do not find any significant difference in XPS analysis between coated and uncoated samples. Then, how can they conclude that they have coated their particles effectively?
Q6: Why do the authors say that the particles are laid in the thickness direction? From SEM images 5e and 5f, we cannot conclude anything about some preference in the position of the particles within the polymer.
3.2. Electrical properties testing results of XLPE/α-Al2O3 nanocomposites
Q7: The authors should include the DC conductivity of XLPE with uncoated Al2O3 particles to check the difference between them in Figure 10. The authors cannot talk of electrical percolation in this study. Their materials are insulating materials.
Q8: Figures 8 and 11 should be put together in order to compare them better. The authors should discuss better the main differences between them. Why do the authors say that the DC breakdown strength of XLPE/uncoated Al2O3 is more dispersive? What is the meaning of more dispersive?
Author Response
Point 1: This paragraph is confuse:
“Direct current breakdown testing, electrical conductivity tests and measurements of space charge indicated that the surface modification of α-Al2O3 improved the direct current voltage breakdown strengths of crosslinked polyethylene/α-Al2O3 nanocomposites”.
The authors should introduce the more important specific values of their study.
Response 1: I have revised and refined this sentence. The most important values of this study includes two aspects, the first was the addition of α-Al2O3 nanosheets improved direct current breakdown strength, reduced the DC conductivity and suppressing space charge accumulation of XLPE, and the second is the improvement of electrical properties of nanocomposites by surface modification of α-Al2O3 nanosheets. I have modified it as follows:“Direct current breakdown testing, electrical conductivity tests and measurements of space charge indicated that the addition of α-Al2O3 nanosheets introduced a large number of deep traps, blocked the charge injection and decreased the charge carrier mobility, thereby significantly reducing the conductivity, improving direct current breakdown strength and suppressing space charge accumulation in the crosslinked polyethylene matrix. Besides, the results of direct current breakdown testing and electrical conductivity tests also showed that the surface modification of α-Al2O3 nanosheets effectively improved the direct current breakdown strength and reduced the conductivity of crosslinked polyethylene/α-Al2O3 nanocomposites.”
Point 2: The authors do not respond to this comment in their last revision (response 4):
The authors state the following:
“Compared to other crystalline forms of α-Al2O3, α-Al2O3 possesses many excellent properties, such as high hardness, high dimensional stability, excellent electrical insulation, low dielectric constant and dielectric loss”
My question is if the aim of the work is to produce materials with higher breakdown voltage, then they should work with insulating polymers and fillers with high dielectric constants and low dielectric losses.
Response 2: I am sorry to omitted this problem in the last revision. Due to my negligence, I wrote it wrong. The sentence essentially was “Compared to other crystalline forms of Al2O3, α-Al2O3 possesses many excellent properties, such as high hardness, high dimensional stability, excellent electrical insulation, high dielectric constant and low dielectric loss.” I have modified it. To prove that, I quote here a reference describing the advantages of α-Al2O3.
Point 3: The authors should introduce the specific values obtained in the references 16-18 to compare them after with the values obtained in this work. The results obtained in this work are better than the results provided by references 16-18 with particles of Al2O3?
Response 3: Only breakdown strengths appear in the references 16 and 17. And I have compared the DC breakdown strengths obtained in this work with the specific values obtained in the references 16 and 17.By comparson, I draw a conclusion that the breakdown field strength (320 kV/mm) of the XLPE/coated α-Al2O3 nanocomposites containing 1.0 wt% coated α-Al2O3 in this work was much higher than the breakdown field strength (200 kV/mm) of sandwich-structured Al2O3-LDPE/LDPE/Al2O3-LDPE nanocomposites in ref 16. And the breakdown field strength (320 kV/mm) was lower than the breakdown strength (450kV /mm) of polyethylene/alumina nanocomposites in ref 17. But the breakdown strength of neat LDPE in ref 17 have already reached 450 kV/mm and the breakdown strength of the pure XLPE in this study was only 220 kV/mm. Therefore, the effect of adding coated α-Al2O3 nanosheets in this work was more significant.
Point 4: The authors must explain how the DC conductivity is calculated. Why did the authors choose samples with different thicknesses for both analysis? My recommendation is to test whenever possible samples with the same dimensions for electrical analysis.
Response 4:
- The method calculating DC conductivity:
I have added the calculating method in the article: The data measured by insulation resistanced tester was resistance (Rv). The resistance of each specimen was measured five times and averaged. The resistivity (ρv)can be expressed as ρv=RvS/h
Where S is the area of the electrode and h is the thickness of the specimen. The DC electrical conductivity was the reciprocal of resistivity.
- About different thickness and diameter for both analysis:
The specimens for breakdown testings should be enough big to be measured about 15 times. To avoid the effects of water and bubbles in the specimens on the breakdown data, the specimens need to be prepared very thin, so the square specimens with a thickness of 0.15 mm and a side length of 11.5 cm were prepared for the high voltage breakdown testings.
However, the specimens for PEA measurements and DC conductivity tests can be preapared small in diameter, because they just need to cover the electrodes. But, the specimens for PEA measurenebt should be thick to better observe the behavior of the inner space charge. So the round specimens with a thickness of 0.3 mm and a diameter of 8 cm were prepared for the DC electric conductivity tests and measurements of space charge.
In the preparation of the nanocomposites, we used the plate vulcanizing machine to prepare the specimens. Even if the quality of sample is same, it is difficult to press into the same thickness of the specimens. I have tried several times, but I can’t get the same thickness. But in DC breakdown testings and conductivity tests, the influence of thickness on them can be ignored by calculating. And in the analysis of space charge in the specimens, we can get the correct conclusions even if the sample thickness is different.
Point 5: The authors do not find any significant difference in XPS analysis between coated and uncoated samples. Then, how can they conclude that they have coated their particles effectively?
Response 5: XRD is a commonly used method for analyzing and testing crystals. The patterns showed that the prepared α-Al2O3 nanosheets have good crystallinity. But the silane layer coated on the surface of α-Al2O3 nanosheets can be shown in the patterns. Because its content is so low, even when it does appear, its peaks are hard to find, small, smooth peaks. So we can’t conclude the success of surface modification by XRD pattern, but we can conclude it by HRTEM images and FTIR spectrum. Because, in the HRTEM images, the α-Al2O3 has distinct lattice, while the silane layer does not. And in the FTIR spectrum, there are many characteristic peaks of organic bonds in silane layer.
Point 6: Why do the authors say that the particles are laid in the thickness direction? From SEM images 5e and 5f, we cannot conclude anything about some preference in the position of the particles within the polymer.
Response 6: About SEM images, the other reviewer aslo raise a question that SEM images of some cross section should be provided. So I have deleted the extra SEM pictures (Fig 5a and 5b) and changed SEM pictures you mentioned. In the new SEM images, Fig 5a is to show the size and uniformity of the α-Al2O3 nanosheets; Fig 5b is to show the thickness and high aspect ratio of α-Al2O3 nanosheets; Fig 5c and 5d are the images of cross section of XLPE/coated α-Al2O3 nanocomposites and XLPE/uncoated α-Al2O3 nanocomposites, It can be seen that in the XLPE/uncoated α-Al2O3 nanocomposites, α-Al2O3 nanosheets aggregated, while in the XLPE/coated α-Al2O3 nanocomposites, the α-Al2O3 nanosheets were dispersed evenly in the XLPE matrix. Fig 5e and 5f show the cross sections of the nanocomposite specimens after liquid nitrogen quenching section. The Fig 5e and 5f can prove that “The paticles are laid in the thickness direction”.
Point 7: The authors should include the DC conductivity of XLPE with uncoated Al2O3 particles to check the difference between them in Figure 10. The authors cannot talk of electrical percolation in this study. Their materials are insulating materials.
Response 7: I tested the DC conductivity of the XLPE/uncoated α-Al2O3 nanocomposites and added it in my article. By comparison, it can be seen that the surface modification of α-Al2O3 nanosheets can also reduce the DC conductivity of XLPE/ α-Al2O3 nanocomposites. And, thanks for your reminding, I realized that the “electrical percolation” is a term used to describe conductive materials, not insulating materials. I have deleted the content that refers to electrical percolation in the article.
Point 8: Figures 8 and 11 should be put together in order to compare them better. The authors should discuss better the main differences between them. Why do the authors say that the DC breakdown strength of XLPE/uncoated Al2O3 is more dispersive? What is the meaning of more dispersive?
Response 8: The other reviewer also mentioned this problem: The DC breakdown strengths of same pure XLPE in two experiments were so different under same conditions, it is very weak to compare them and draw a conclusion. To solve this, I tested all the specimens again, including the one pure XLPE specimen, four specimens of XLPE/uncoated α-Al2O3 nanocomposites containing 0.2 wt%, 0.5 wt%, 1.0 wt%, and 2.0 wt% uncoated α-Al2O3 and four specimens of XLPE/coated α-Al2O3 nanocomposites containing 0.2 wt%, 0.5 wt%, 1.0 wt%, and 2.0 wt% coated α-Al2O3. To improve the reliability of the data, I conducted 15 times of breakdown experiments on each of 9 specimens and chose the middle 10 data among the 15 data according the size to carry out data processing. For better comparision, I put the two figures of DC voltage breakdown tests and two sets of date of DC condutivity tests together. Finally, by comparision, the conclusion can be drawn convincingly that the surface modification of α-Al2O3 nanosheets can improve the DC breakdown strengths and reduce the DC conductivity of XLPE/α-Al2O3 nanocomposites.
Thanks so much for all your valued comments! If there are any other problems in my article, please raise them and I will seriously correct them in time.

Round 3
Reviewer 1 Report
The revision was conducted according to the reviewer comments and the article can now be accepted for publication.
Author Response
Thanks for your professional guidance, so that I can recognize my shortcomings in this work and correct them in time.
Thanks!
Kind regards,
Xiangjin Guo
Reviewer 2 Report
The discussion and evidence of final strengths were improved in the last version.
I have some additional questions/ comments for the authors, after reading the last version and responses from the authors before acceptance for publishing.
Abstract
The abstract is still confusing. Their coated particles reduce the conductivity of XLPE, and improved direct current breakdown strength, etc. when compared with uncoated particles in neat XLPE.
The authors should introduce in the abstract the more important specific values of their study (320 kV/mm), etc.
- Results and Discussion
3.1. Characterization results of α-Al2O3 nanosheets
Q1. The caption text in new Fig. 5 must indicate the content of particles in XPLE. Why are the Fig. 5e and 5f blurred with respect to the others? We cannot conclude from Fig. 5c and 5d the presence of agglomerates. What are the contents of particles in 5c and 5d? Based on the data provided in the work, we cannot conclude that there is agglomeration of coated and uncoated particles in XPLE.
Q2. The authors claim that the particles are aligned in the XLPE. What is the mechanism during processing that causes that alignment?
3.2. Electrical properties testing results of XLPE/α-Al2O3 nanocomposites
Q3. Authors must put Figure 11 and its corresponding discussion first and then Fig. 8 and 9 together in only one Figure to compare them better. Authors should explain in Figure 11 why the coated and uncoated particles decrease the conductivity of neat XLPE (around 3.25 x 10-13 S/m) to 1.75 x 10-13 for 0.5 % of particles. What is the conductivity mechanism in pure XLPE, what is the conductivity mechanism after adding some insulating particles in XLPE? Are the Al2O3 particles insulating? Can the authors provide the neat conductivity of their particles? Is it possible to make a kind of buckypaper of the particles only and check their conductivity?
Author Response
Response to Reviewer 2 Comments
Point 1: The caption text in new Fig. 5 must indicate the content of particles in XPLE. Why are the Fig. 5e and 5f blurred with respect to the others? We cannot conclude from Fig. 5c and 5d the presence of agglomerates. What are the contents of particles in 5c and 5d? Based on the data provided in the work, we cannot conclude that there is agglomeration of coated and uncoated particles in XPLE.
Response 1: I have added the contents of particles in the caption text of new Fig 5. Fig 5c and 5d are the cross section of XLPE/coated α-Al2O3 nanocomposites and XLPE/uncoated α-Al2O3 nanocomposites containing 2.0 wt% α-Al2O3 nanosheets; Fig 5e and 5f are the cross sections of the nanocomposite specimens containing 1.0 wt% and 2.0 wt% α-Al2O3 nanosheets.
Due to various reasons, Fig 5e and 5f were obtained by other types of scanning electron microscopes (SEM; Phenom ProX) and others were observed by scanning electron microscopy (SEM; FEI·Nova·Nano·SEM450). So the Fig 5e and 5f blurred with respect to the others and I also added this statement to the article.
The contents of particles in Fig 5c and 5d respectively are 1.0 wt% and 2.0 wt%.
About the agglomeration of particles in XLPE. After your repeated reminding and my careful consideration, I realized that the non-uniform dispersion of nanoparticles in the XLPE could not be called agglomeration of nanoparticles. It can only be described as the non-uniform dispersion of α-Al2O3 nanosheets in the XLPE matrix. I have made changes in the article.
Point 2: The authors claim that the particles are aligned in the XLPE. What is the mechanism during processing that causes that alignment?
Response 2: In the preparation of XLPE/α-Al2O3 nanocomposites, I used three curing presses to press the specimens. The first curing press was to melt the nanocomposites, the second was to crosslink the LDPE to XLPE and the third was to cool them rapidly. The specimens used in the work were thin enough. During the crosslink, at high temperature (180℃) and high pressures (16MPa), the XLPE polymer matrix rapidly melted. so the α-Al2O3 nanosheets with high hardness and high aspect ratioin in melting XLPE will be distributed in the most stable states. As an analogy, it’s just like many disc-shaped stones are laid in the water. The metaphor may not be appropriate, but I can’t describe it better.
Point 3: Authors must put Figure 11 and its corresponding discussion first and then Fig. 8 and 9 together in only one Figure to compare them better. Authors should explain in Figure 11 why the coated and uncoated particles decrease the conductivity of neat XLPE (around 3.25 x 10-13S/m) to 1.75 x 10-13 for 0.5 % of particles. What is the conductivity mechanism in pure XLPE, what is the conductivity mechanism after adding some insulating particles in XLPE? Are the Al2O3 particles insulating? Can the authors provide the neat conductivity of their particles? Is it possible to make a kind of buckypaper of the particles only and check their conductivity?
Response 3:For the sake of explanation, I have put Figure 11 and its corresponding discussion together and put Fig 8 and 9 together in only one Figure to compare them better.
I have added the conductivity mechanism of pure XLPE and the conductivity mechanism of XLPE/α-Al2O3 nanocomposites after adding some insulating particles in XLPE. Under the high voltage, the XLPE will break down and become from a insulator to a conductor, but before the breakdown, the XLPE is not an absolutely non-conductive insulator, and a weak current will appear in the material. Because there are usually only a few free electrons in the insulating material, the charges mainly come from the intrinsic ions and impurity particles. The addition of α-Al2O3 nanosheets introduce a large number of traps, the number of traps in the nanocomposites is much larger than that in the pure XLPE, which reduces the mobility of carriers in the nanocomposites and prevent the injection of charges, thus reducing the conductivity of the nanocomposites.
The α-Al2O3 is an inorganic insulating materials, but we can’t obtain and provide its neat conductivity, even if press it to a thin piece. About measurement of it, I have tried and failed. Because its conductivity is so small that we can’t test it with our relevant instruments.
Point 4: The authors should introduce in the abstract the more important specific values of their study (320 kV/mm), etc.
Response 4: I have not understand your meanings in the previous two revisions. After your reminding of specific numbers, I have realized this problem and described the important specific values of this work in the Abstract.
Thanks for your professional guidance, so that I can recognize my shortcomings in this work and correct them in time. I sincerely admire your patience and academic knowledge.
Thanks!
Kind regards,
Xiangjin Guo
